# Does Vitamin B6 Act as an Exercise Mimetic in Skeletal Muscle?

**DOI:** 10.3390/ijms25189962

**Published:** 2024-09-15

**Authors:** Norihisa Kato, Yongshou Yang, Chanikan Bumrungkit, Thanutchaporn Kumrungsee

**Affiliations:** 1Graduate School of Integrated Sciences for Life, Hiroshima University, Higashi-Hiroshima 739-8528, Japan; m235016@hiroshima-u.ac.jp; 2School of Life Sciences, Anhui University, Hefei 230601, China; 21193@ahu.edu.cn; 3Smart Agriculture, Graduate School of Innovation and Practice for Smart Society, Hiroshima University, Higashi-Hiroshima 739-8527, Japan

**Keywords:** vitamin B6, pyridoxal 5′-phosphate, exercise, skeletal muscle, myogenesis, inflammation, NLRP3 inflammasome, PLP enzyme, H_2_S, AMPK

## Abstract

Marginal vitamin B6 (B6) deficiency is common in various segments worldwide. In a super-aged society, sarcopenia is a major concern and has gained significant research attention focused on healthy aging. To date, the primary interventions for sarcopenia have been physical exercise therapy. Recent evidence suggests that inadequate B6 status is associated with an increased risk of sarcopenia and mortality among older adults. Our previous study showed that B6 supplementation to a marginal B6-deficient diet up-regulated the expression of various exercise-induced genes in the skeletal muscle of rodents. Notably, a supplemental B6-to-B6-deficient diet stimulates satellite cell-mediated myogenesis in rodents, mirroring the effects of physical exercise. These findings suggest the potential role of B6 as an exercise-mimetic nutrient in skeletal muscle. To test this hypothesis, we reviewed relevant literature and compared the roles of B6 and exercise in muscles. Here, we provide several pieces of evidence supporting this hypothesis and discuss the potential mechanisms behind the similarities between the effects of B6 and exercise on muscle. This research, for the first time, provides insight into the exercise-mimetic roles of B6 in skeletal muscle.

## 1. Introduction

Vitamin B6 (B6) is a water-soluble vitamin found in various foods such as fish, whole grains, and bananas. There are six isoforms of B6, known as B6 vitamers, which include pyridoxine (PN), pyridoxal (PL), pyridoxamine (PM), and their phosphorylated forms. Among these, pyridoxal 5′-phosphate (PLP) is the most active and acts as a coenzyme in over 150 reactions. These include the synthesis, transformation, and degradation of amino acids; the provision of one-carbon units; trans-sulfation; synthesis of tetrapyrrolic compounds, polyamines, and hydrogen sulfide (H_2_S); biosynthesis and degradation of neurotransmitters; as well as glycogen breakdown [1,2]. Additionally, B6 itself has radical scavenging activity [1,2].

While severe B6 deficiency (plasma PLP: < 20 nmol/L) is rare, marginal B6 deficiency (suboptimal levels or slight deficiency, plasma PLP: 20–30 nmol/L) appears to be more common worldwide [3,4]. In developed countries such as Germany, the United Kingdom, and the Netherlands, 5–25% of the male population does not meet the recommended B6 intakes (1.2–1.5 mg/day for men and women from 19 to 50 years old) [3]. Moreover, B6 deficiency seems more pronounced in older adults [5,6,7]. There is increasing evidence suggesting that B6 exerts protective effects against chronic diseases such as cardiovascular diseases (CVD) and cancers by reducing inflammation, inflammasomes, and oxidative stress [1,2].

Over recent decades, there has been a noticeable and rapid aging of the global population. In societies where aging is particularly pronounced, sarcopenia, a condition characterized by the loss of muscle mass and strength with aging, has become a significant concern in research focused on maintaining physical function and metabolic health [8]. The most prevalent protective approach for sarcopenia is physical exercise therapy. While studies have explored the impact of nutrients such as protein, omega-3 polyunsaturated fatty acids, and vitamin D on the progression of sarcopenia in older adults [9,10], research on the effects of other nutrients remains limited.

Several studies have highlighted the inverse association between B6 status and the risks of sarcopenia and mortality in older adults [11]. Conversely, few studies have indicated a significant association with other vitamins, such as vitamins A, B1, B2, B3, B9, B12, C, D, and E [11]. An early study by Suboticanec et al. indicated that B6 supplementation in B6-deficient young individuals (aged 12–14 years) significantly improved B6 status and enhanced physical fitness assessed through bicycle ergometer tests [12]. These findings led us to a subsequent study on the effects of dietary B6 levels on gene expression in the skeletal muscle of rats [13]. The results indicated that supplemental B6 increased the expression of various genes involved in myogenesis, cytoprotection, anti-oxidation, and mitochondrial biogenesis. Notably, such expressions are increased by exercise as well [13].

Moreover, a recent study by Komaru et al. revealed that supplemental B6 to a B6-deficient diet promotes satellite cell-mediated myogenesis in the skeletal muscle of mice [14], similar to the effects of exercise. Based on this evidence, we hypothesized that supplemental B6 possesses exercise-mimetic properties in skeletal muscle (Figure 1). To test this hypothesis, in this study, we reviewed relevant literature from databases like PubMed, Scopus, and Google Scholar, spanning publications from 1970 onward with the following keywords: “vitamin B6”, “PLP”, “exercise”, “inflammasome”, “inflammation”, “oxidative stress”, “skeletal muscle”, “myogenesis”, “glycogenolysis”, “gluconeogenesis”, “mitochondria”, “cytoprotection”, “H_2_S”, “AMPK”, and “the kynurenine pathway”. Additionally, we discussed the underlying mechanisms through which B6 may exert exercise-mimetic effects. 

## 2. Muscle Protein Synthesis and Gene Expression

Sampson et al. [15] investigated the effects of consuming diets with different levels of B6 (a marginal B6-deficient diet [1.2 mg PN HCl/kg diet] versus a normal diet [5.8 mg PN HCl/kg]) over nine weeks of the protein fractional synthesis rate in the skeletal muscle of growing rats. Compared to the marginal B6-deficient diet, the normal diet tended to increase protein synthesis rates (+26%, *p* = 0.05) [15]. In another study, Suidasari et al. [13] compared the effects of consuming a marginal B6-deficient diet (1 mg PN HCl/kg) versus a recommended level of B6-containing diet (7 mg PN HCl/kg) over six weeks on gene expression in rat skeletal muscle. They observed significant increases in the expression of various genes responsible for myokines, myogenesis, mitochondrial biogenesis, and energy metabolism in the higher B6 group [13]. The products of these genes included interleukin (IL)-6 (a stimulator of muscle hypertrophy) [16], IL-7 (a stimulator of immune function) [16], IL-8 (a stimulator of muscle neovasculization) [16], secreted protein acidic and rich in cysteine (SPARC, an essential factor for muscle development and regeneration, mitochondrial biogenesis, and 5′ adenosine monophosphate-activated protein kinase (AMPK) activator) [17], growth differentiation factor 11 (GDF-11, a factor for the rescue of the proliferative and regenerative capabilities of muscle) [18], myonectin (an activator of AMPK, mitochondrial biogenesis and stimulator of muscle growth) [19], leukemia inhibitory factor (LIF, a key regulator of muscle growth and regeneration) [20], apelin (a potential regulator of physical performance) [21], retinoic acid receptor responder 1 (RARRES1, a tumor suppressor), [22] heat shock protein 60 (HSP60, a stimulator of muscle cytoprotection and mitochondrial biogenesis) [23] and myogenin (an essential regulator of myofibre growth and muscle stem cell homeostasis) [24]. Additionally, the expression of antioxidant genes of nuclear factor erythroid 2-related factor 2 (Nrf2) [25] and its regulated enzymes involving heme oxygenase 1 (HO-1) [26], superoxide dismutase 2 (SOD-2), glutathione peroxidase 1 (GPx-1), and glutathione S-transferase (GST) were significantly higher in the 7 mg B6/kg group [13]. While the effect of exercise on the expression of RARRES1 gene is unclear, exercise training is known to up-regulate the expression of all other genes mentioned, similar to the effects observed with B6 supplementation [13,16,27,28,29,30]. These findings have led to a hypothesis suggesting that B6 may play a role in skeletal muscles as an exercise-mimetic factor (Figure 1).

## 3. Satellite Cell-Mediated Myogenesis

Satellite cells in skeletal muscle play a crucial role in maintaining muscle homeostasis and facilitating repair processes, yet they are vulnerable to damage from inflammation and oxidative stress [20]. Komaru et al. [14] conducted a study demonstrating that a marginal B6-deficient diet (1 mg PN HCl/kg) led to a decrease in satellite cell numbers in mouse skeletal muscle compared to a diet with adequate B6 levels (35 mg PN HCl/kg). This deficiency resulted in impaired proliferation and self-renewal of satellite cells during myogenesis. Moreover, Kumar et al. highlighted the significance of PL, a key metabolite of B6, which was significantly up-regulated in differentiating human myoblasts, suggesting the role of B6 in myogenic progression [31]. Additionally, supplemental B6 stimulates the synthesis of carnosine, an ergogenic dipeptide (β-alanyl-L-histidine) known for its presence in Type II glycolytic muscle fibers with contractile properties [32,33]. A study by Liu et al. has suggested that carnosine may promote muscle growth by stimulating the proliferation and satellite cells through the Akt/mTOR/S6K signaling pathway [34].

Notably, the increased proliferation and self-renewal of satellite cells observed with B6 supplementation mirror effects seen with endurance exercise [30,35] (Figure 1, Table 1). Similarly, B6 supplementation increases the expression of genes involved in myogenesis in rat skeletal muscle [13], similar to the effect of exercise. Therefore, supplemental B6 may exert exercise-mimetic roles in skeletal muscle.

## 4. Inflammation and Inflammasome

The P2X7 receptor (P2X7R)-nucleotide-binding oligomerization domain-like receptor family pyrin domain-containing 3 (NLRP3) inflammasome signaling pathway has emerged as a potential target for treating muscle dystrophy [36,37]. P2X7R is a purinergic receptor activated by ATP released during cellular injury and is expressed in muscle satellite cells. The NLRP3 inflammasome facilitates the processing of cytokines such as IL-1β and contributes to inflammatory diseases, including skeletal muscle disorders, which are often accompanied by oxidative stress. Earlier studies have linked skeletal myositis to NF-κB signaling mechanisms [38] and observed that B6 inhibits this pathway [39] (Figure 1, Table 1). Notably, PLP inhibits NLRP3-dependent caspase-1 processing and IL-1β production, suppressing the NLRP3 inflammasome in experiments with LPS-stimulated macrophages and mice treated with LPS [40] (Figure 1, Table 1). Additionally, PLP acts as an antagonist to P2X7R [41]. 

Regular exercise effectively reduces inflammatory cytokine levels associated with NLPR3 inflammasome activation in older adults [42]. Activation of the NLRP3 inflammasome can be inhibited by H_2_S, known for its antioxidant, anti-inflammatory, and myogenic properties [43,44]. H_2_S is produced by cystathionine-γ-lyase (CGL), an enzyme dependent on PLP. Wu et al. demonstrated that treadmill running increases H_2_S concentration and CGL protein expression in the skeletal muscles of rats fed with a high-fat diet [45]. Therefore, higher B6 levels and exercise may increase H_2_S production and reduce inflammasome activation, thereby protecting skeletal muscle (Figure 1, Table 1). 

Recent evidence suggests that elements of the kynurenine pathway, a tryptophan metabolite pathway, influence skeletal muscle formation, metabolism, and function [46]. Some components, such as kynurenine, kynurenic acid, quinolinic acid, and tryptophan metabolites, have been associated with both beneficial and detrimental effects on muscle health, impacting the pathogenesis of conditions like sarcopenia and frailty. PLP-dependent enzymes are involved in the synthesis of these kynurenine pathway metabolites. 

Inflammation and metabolic disorders resulting from NLRP3 inflammasome activation via G protein-coupled receptor 35 (GPR35) can be mitigated by kynurenic acid, synthesized from kynurenine by the PLP enzyme kynurenine aminotransferase (KAT) [47] (Table 1). Schlittler et al. reported increased expression of KAT protein in the muscles of endurance-trained individuals compared to untrained participants [48]. Therefore, supplemental B6 and exercise may prevent muscle atrophy by modulating NF-κB signaling, NLRP3 inflammasome activation, H_2_S signaling, and the kynurenine pathway (Figure 1, Table 1).

## 5. Energy Utilization

Alanine aminotransferase (ALT) and aspartate aminotransferase (AST) are PLP-dependent enzymes. Therefore, they are susceptible to B6 deficiency [49]. ALT converts L-alanine and α-ketoglutarate to pyruvate and glutamate, while AST converts L-aspartate and α-ketoglutarate to oxaloacetate and glutamate in various tissues, including skeletal muscle. These reactions produce pyruvate and oxaloacetate, which are crucial for gluconeogenesis in the liver, providing glucose for muscle function. Low levels of ALT and AST may impair these vital metabolic processes, such as amino acid metabolisms and gluconeogenesis, potentially affecting physical activity (Figure 1, Table 1).

Traditionally, elevated ALT and AST levels in plasma have been indicators of liver damage, but intense exercise and weightlifting can increase these enzymes without liver damage [50]. Field hockey players have also shown increased ALT and AST levels [51], possibly leading to increased amino acid catabolism and gluconeogenesis to support muscle activity. However, the exact mechanism underlying the exercise-induced elevation of these enzymes requires further examination. 

PLP serves as a cofactor for glycogen phosphorylase (GP), an enzyme crucial for muscle glycogenolysis, which releases glucose-1-phosphate from glycogen to provide additional glucose during exercise [52]. This enzyme activity is post-translationally activated during intense muscle contractions [53]. Okada et al. reported that B6 deficiency reduces GP activity in rat skeletal muscle [54]. Therefore, both muscle contraction and B6 supplementation increase GP activity, which could enhance muscle glycogen utilization, potentially improving physical performance and exercise capacity (Figure 1, Table 1).

Sarcopenia, characterized by mitochondrial dysfunction in skeletal muscle, can be alleviated by exercise [55], which enhances the activity of 5-aminolevulinate synthase (ALS, mitochondrial heme biosynthesis enzyme) for increasing cytochromes and other mitochondrial enzymes [56]. ALS requires PLP, and B6 deficiency is associated with mitochondrial dysfunction due to reduced ALS activity [57,58]. A study by Suidassari et al. [13] has shown that B6 supplementation to a B6-deficient diet increases muscle gene expression related to mitochondrial biogenesis and functions, including SPARC [17], myonectin [19], HSP60 [23], Nrf2 [25], and HO-1 [26], as mentioned above. Furthermore, Ciapaite et al. indicate that PLP depletion in pyridoxal 5′-phosphate homeostasis protein (PLPHP)-deficient skin fibroblasts impairs mitochondrial oxidative function [59], suggesting a role for B6 in mitochondrial function (Figure 1). 

AMPK, a marker of cellular energy status, promotes mitochondrial biogenesis [60]. Shan et al. demonstrated that exposure to PL increases AMPK Thr172 phosphorylation in cultured primary macrophages in a time- and dose-dependent manner [61]. Therefore, B6 may enhance mitochondrial biogenesis via AMPK activation. Moreover, sarcopenia is inhibited by the regulation of the AMPK/Sirt1 pathway induced by resveratrol and exercise in aged rats [55]. AMPK-dependent pathways also down-regulate NLRP3 inflammasome activation during aging [62]. Therefore, exercise-induced AMPK activation may constitute a therapeutic strategy for chronic inflammatory myopathy [63]. Accordingly, B6 and exercise may play a role in protecting against musculoskeletal aging by regulating NLRP3 inflammasomes, AMPK signaling, and H_2_S signaling (Figure 1, Table 1).

## 6. Relationship between Exercise and B6 Metabolism

As mentioned above, there are several similarities between the roles of supplemental B6 and exercise in skeletal muscle. This raises the question of the underlying mechanisms behind these similarities. We postulated that exercise training might modulate B6 metabolism. To test this possibility, we surveyed past literature and observed seven clinical studies [64,65,66,67,68,69,70] and three rat studies [71,72,73] (Table 2) in which either acute or prolonged exercise training significantly increased B6 status, as evidenced by increased PLP and B6 metabolites. However, a study conducted by Gaume et al. found no significant effect on plasma PLP levels in rats after 8 weeks of swim training [74]. Notably, Deiana et al. compared the metabolomic profile by analyzing runners’ sera before and after a half marathon, finding that metabolites involved in B6 salvage, such as PLP and PMP, were significantly increased in runners (Table 2) [70]. Furthermore, a recent study by Alme et al. showed a significant association between sedentary time and the PAr index (4-PA: [PL + PLP], an indicator of B6 catabolism) in patients (r^2^ = 0.37, *p* < 0.001) [75]. Therefore, exercise training may frequently increase B6 status by reducing B6 catabolism. Based on these studies, we speculate that increased B6 status might be at least partially involved in the underlying mechanism of the effects of exercise (Figure 1). This could explain why dietary B6 exerts similar effects to exercise on skeletal muscle.

Moreover, exercise treatment induces several PLP enzymes, such as ALT, AST, CGL, KAT, and ALS, as mentioned above. These effects on PLP enzymes might, in part, relate to increased levels of PLP. It is also worthwhile to mention that exercise induces other PLP enzymes such as ornithine decarboxylase (ODC) [76], histidine decarboxylase (HDC) [77], and serine palmitoyltransferase (SPT) [78], which play roles in the adaptation to exercise. Emerging metabolome studies have reported alterations in several metabolic pathways, including amino acid metabolisms, due to exercise [79]. Hence, it is of interest to reveal the responses of a variety of PLP enzymes to exercise.

Two mechanisms by which exercise increases B6 levels were considered. First, moderate-intensity exercise can exert an antioxidant effect [80,81]. Pyridoxal kinase (PK) is typically susceptible to oxidative stress [82]. Since exercise training can suppress oxidative stress, the reduced oxidative stress from exercise might protect PK from oxidation, thereby sustaining PLP levels. Second, catabolism of B6 to 4-PA increases during systemic inflammation [83]. Blood PLP is dephosphorylated by tissue-nonspecific alkaline phosphatase (TNAP), which is induced by inflammatory mediators such as TNFα and IL-1β [84]. Increased TNAP activity during inflammation might be partially responsible for higher B6 catabolism. Therefore, a reduction in inflammation through exercise might contribute to elevated B6 levels by lowering TNAP activity. However, further study is necessary to confirm these possibilities. 

Overall, the similarity between the roles of exercise and B6 supplementation in a B6-deficient diet in skeletal muscle could be partially explained by increased B6 levels (Figure 1). This hypothesis has yet to be fully validated. Nevertheless, we may suppose that increased PLP levels might be an adaptive response to physical training.

## 7. Relationship between Exercise and Carnosine Metabolism

As mentioned above, supplemental B6 to a B6-deficient diet increased the levels of L-carnosine, an ergogenic factor, in skeletal muscles and heart [32,85]. Interestingly, β-alanine (BA), a precursor of carnosine, significantly increased by B6 supplementation, while ornithine sharply decreased. Ornithine can be metabolized to polyamines by ODC (a PLP enzyme), which then can be converted to BA. Therefore, B6 intake could enhance the conversion of ornithine to carnosine, possibly via increased ODC activity [32]. Carnosine has various beneficial properties, including its effects on muscle contraction and its antioxidant, anti-inflammatory, anti-glycation, anti-aging, pH buffering, glycogen phosphorylase-activating, and mitochondrial function-promoting properties [86,87]. Therefore, increased carnosine levels resulting from supplemental B6 may enhance physical activity and improve muscle function in older adults. 

We have postulated that supplemental B6 could mimic the potential role of exercise in skeletal muscle. If correct, exercise training may increase muscle carnosine levels similarly to B6 supplementation. Supporting this hypothesis, six studies [88,89,90,91,92,93] reported that exercise significantly increased muscle carnosine levels, while three studies [94,95,96] did not show a significant effect (Table 3). A study by Hoetker et al. indicated that a single session of high-intensity interval training (HIIT) significantly increased muscle carnosine, but a further 6 weeks of HIIT significantly decreased carnosine level [97] (Table 3). These mixed results suggest that further research is needed to determine the optimal intensity and duration of physical exercise, as well as the types of sports that effectively increase carnosine levels. 

In addition to changes in muscle carnosine levels due to exercise, changes in carnosine levels in response to acute muscle injuries have been reported [98]. In a human muscle micro-dialysis study [98], the insertion of a micro-dialysis probe, which can potentially cause muscle cell damage, induced a marked increase in interstitial carnosine levels 1 h post-insertion, but basal levels were restored within 2 h post-insertion. Exercise generally induces muscle injury, which leads to muscle regeneration and hypertrophy. The mobilization of carnosine during muscle injury suggests a possible role for carnosine in regulating muscle recovery or regeneration after exercise. Interestingly, carnosine content significantly declines in sarcopenic or aging muscles, which are underused or show low levels of exercise [99].

In addition to carnosine, a previous study reported that physical training increases polyamine (spermidine and spermine) levels and ODC activity in rat skeletal muscle [100]. ODC is responsible for the biosynthesis of polyamines as well as carnosine. Polyamines are essential for cell growth and differentiation and possess antioxidant and anti-inflammatory effects [101]. Polyamine levels were significantly lower in the skeletal muscles of aged animals [102]. Therefore, supplementation with spermine and spermidine may help prevent or treat muscle atrophy. The increased activity of ODC by exercise and B6 supplementation might play a crucial role in the adaptation of skeletal muscle to exercise.

## 8. Conclusions and Further Directions

In this article, we reviewed past literature to test our hypothesis that supplemental B6 to a B6-deficient diet plays exercise-mimetic roles in skeletal muscles. Overall, accumulating evidence supports this hypothesis by highlighting the similarities between the roles of supplemental B6 and physical training in skeletal muscle, including gluconeogenesis, glycogenolysis, inflammasome, inflammation, oxidative stress, myogenesis, mitochondrial biogenesis, the kynurenine pathway, H_2_S signaling, and AMPK signaling (Figure 1, Table 1). Furthermore, previous studies have shown that regular exercise frequently increases B6 levels in humans and rodents (Table 2). Accordingly, increased B6 levels might be partially involved in the underlying mechanisms of the beneficial effects of exercise. This article suggests that B6 exerts similar roles to those of exercise in improving musculoskeletal function, indicating that it may represent an exercise-mimetic nutrient in skeletal muscle. This study provides valuable insight into the roles of B6 and exercise by comparing emerging knowledge from each area. Further study is of great interest to investigate the exercise-mimetic roles of B6 in several chronic age-related disorders such as CVD, diabetes, and cancers [1,2]. To the best of our knowledge, this is the first study providing insight into the exercise-mimetic roles of B6 in skeletal muscle.

## Figures and Tables

**Figure 1 ijms-25-09962-f001:**
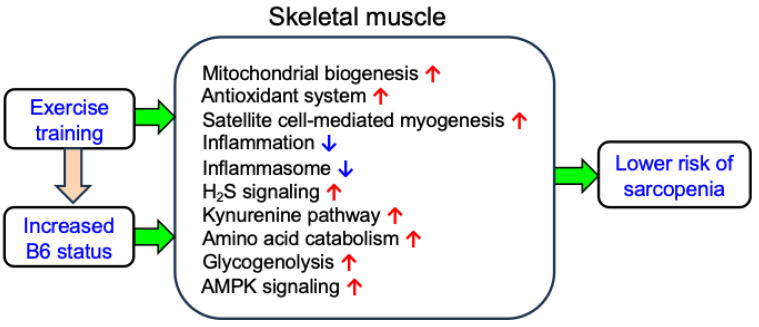
Potential roles of B6 as an exercise-mimetic in skeletal muscle. **↑**: up-regulation, **↓**: down-regulation.

**Table 1 ijms-25-09962-t001:** Comparison of the effects of B6 versus exercise on skeletal muscle.

Effects of B6 and Exercise on Skeletal Muscle	+B6	+Exercise
mitochondrial biogenesis	Up *	Up
antioxidant system	Up	Up
satellite cell-mediated myogenesis	Up	Up
NF-*κ*B dependent inflammation	Down **	Down
NLRP3 inflammasome	Down	Down
H_2_S signaling	Up	Up
the kynurenine pathway (kynurenic acid production)	Up	Up
amino acid catabolism via ALT and AST	Up	Up
glycogenolysis via glycogen phosphorylase	Up	Up
AMPK signaling	Up	Up

* Up-regulation, ** Down-regulation.

**Table 2 ijms-25-09962-t002:** Clinical and animal studies on the effects of exercise training on B6 status.

Study (Year)	Subjects/ Animals	Training Method	Key Results	B6 Levels
Leklem et al. (1983) [64]	20 male adolescent trained athletes	Plasma PLP was determined in the athletes before and immediately after a 4500-m run (total 3 runs, *n* = 6, 7, and 7).	Plasma PLP and total B6 (PLP + PL + PM + PN) increased significantly after long distance runs. Urinary 4-pyridoxic acid (4-PA) was unaffected.	Up *
Manore et al. (1987) [65]	5 young-trained, 5 young-untrained, and 5 postmenopausal-untrained women	After women daily exercised (80% VO_2_ max; 20 min) for 2 weeks, plasma PLP was determined in the women pre- and post-exercise on a cycle ergometer.	Plasma PLP, total B6 (PLP + PL + PM + PN), and urinary 4-PA significantly increased after exercise.	Up
Manore et al. (1988) [66]	5 young-trained, 5 young-untrained, and 5 postmenopausal-untrained women	Subjects were fed four diets (varying carbohydrate and B6 levels) for 7 weeks period. Subjects were exercised at the end of each dietary period at 80% VO_2_ max for 20 min on a cycle ergometer.	Plasma PLP levels significantly increased from pre- to post-exercise. From post- to post-60 min of exercise, plasma PLP significantly decreased. There was no difference in PLP levels for dietary groups.	Up
Hofmann et al. (1991) [67]	6 men (30 ± 2 year-old)	Subjects ran at 60–65% of VO_2_ max for 2 h. Plasma PLP during exercise was determined.	Plasma PLP levels significantly increased during exercise. There was no difference in the effects of high vs. moderate exercise intensity on the PLP level.	Up
Crozier et al. (1994) [68]	physically active 5 women and 4 men (18–35 year-old)	Subjects were tested twice at 60% and 85% of VO_2_ max for 20 and 30 min on a bicycle ergometer. Blood samples were obtained before and during exercise.	Prolonged aerobic exercise induced a consistent rise in plasma PLP. With 79% of the rise in PLP concentration within 5 min.	Up
Hadj-Saad et al. (1995) [71]	40 male SD rats (160–180 g)	Rats were subjected to acute force-swimming for 0, 1, 1.5, 2, and 2.5 h (*n* = 8, respectively).	Plasma PLP, total B6, 4-PA, gastrocnemius muscle PLP, liver PLP, PL, PM, total B6, and kidney PM were increased with swimming length.	Up
Hadj-Saad et al. (1997) [72]	36 male SD rats (160–180 g)	Rats were subjected to a swimming program, which consisted of swimming up to 1 h/d, 6 d/week, for a total of 9 weeks (*n* = 6).	In gastrocnemius muscle, PLP, PMP, and total B6 levels were higher in trained rats than in sedentary rats. Levels of PM, PLP, and PL in the liver were higher in the trained rats. Plasma PLP levels were unaffected.	Up
Okada et al. (2001) [73]	40 male Wistar rats (3 week-old)	Rats were fed a B6-restricted diet (1.5 mg PN HCl/kg) for 5 weeks. A swimming program for the exercise group (*n* = 8) consisted of swimming up to 1 h every day for 3 weeks (*n* = 7 for the control group).	Exercise treatment significantly increased liver PLP and PMP, kidney total B6, and muscle PMP. Plasma PLP levels were unaffected.	Up
Gaume et al. (2005) [74]	16 male SD rats (160–180 g)	Exercise consisted in swimming up to 1 h/d, 5 d/week, for a total of 8 weeks (*n* = 8). Exercise intensity was increased every week (from 10 min to 1 h).	There was no difference in plasma PLP concentrations between the training group and the non-training group.	No change **
Venta et al. (2009) [69]	29 male aerobic athletes (14–22 year-old)	Subjects were examined before and 30 ± 5 min after a specific test to exhaustion during a low-intensity training period.	Plasma PLP concentrations increased significantly after acute exercise.	Up
Deiana et al. (2019) [70]	6 male amateur runners (40 ± 8 year-old)	The runners carried out a 21.1 km half marathon. Sera were obtained before and after the half marathon.	Serum PLP and PMP levels were significantly increased following a half marathon performance.	Up

* Significant change (*p* < 0.05), ** No significant change (*p* > 0.05).

**Table 3 ijms-25-09962-t003:** Studies on the effects of exercise training on carnosine levels.

Study (Year)	Subjects/ Animals	Training Methods	Key Results	Carnosine Levels
Dunnett et al. (2002) [88]	thoroughbred horses (4 geldings, 2 fillies; 5–13 year-old)	For 4 weeks, horses were exercised continuously at an intensity that was increased stepwise to a point of near-maximal performance. Blood samples were collected pre-exercise and 30 min post-exercise.	A significant increase in plasma carnosine levels after intense exercise.	Up *
Suzuki et al. (2004) [89]	6 healthy male university students (22 ± 2 year-old)	Students were trained 2 d per week for 8 weeks on an electronic cycle ergometer. During the first 2 weeks, a single bout of sprinting was performed. In the last 6 weeks of sprinting were performed with 20 min rest intervals.	There was a significant increase in carnosine concentration in skeletal muscle following training.	Up
Kendrick et al. (2009) [94]	14 male students	Seven male students (22 ± 3 year-old) were supplemented with β-alanine (BA) for 4 weeks (6.4 g/d) and seven with a matching placebo (22 ± 3 year-old). Subjects undertook 4 weeks of isokinetic training with the right leg (T) while the left leg was untrained (UT), serving as a control.	There was no significant change in the carnosine content in either the T or UT legs with a placebo.	No change **
Baguet et al. (2011) [95]	20 healthy non-vegetarian students (11 males and 9 females)	Subjects participated in a 5 weeks sprint training intervention (2–3 times per week). They were randomized into a vegetarian (21 ± 1 year-old) and mixed diet (22 ± 2 year-old) groups.	Sprint training had no effect on the muscle carnosine content (pre vs. post-exercise). There was a significant diet × training interaction in soleus carnosine content.	No change
Bex et al. (2014) [90]	young participants (10 nonathletes, 10 cyclists, 10 swimmers, and 5 kayakers)	Participants were supplemented with BA for 23 d. All the athletes were well-trained at baseline (recreational and regional class athletes) and trained at least 8 h/week in their specific sports during the 23 d, whereas the nonathletes were inactive throughout.	Carnosine loading by BA was more pronounced in trained vs. untrained muscles (nonathletes).	Up
Bex et al. (2015) [91]	28 men volunteered to participate	All participants were supplemented with BA for 23 d. The subjects were allocated to a control group (22 ± 2 year-old), HV (22 ± 2 year-old), or HI (22 ± 2 year-old) training group. During the BA supplementation period, the training groups performed nine exercise sessions, consisting of either 75–90 min continuous cycling at 35–45% Wmax (HV) or 3 to 5 repeats of 30 s cycling at 165% Wmax with 4 min recovery (HI).	There was a significant cumulative effect of HV and HI exercise on carnosine accumulation by BA supplementation. However, there was no additional difference between the HV and HI training groups.	Up
Carvalho et al. (2018) [96]	28 cyclists (36 ± 6 year-old)	This study examined the levels of carnosine following high-intensity intermittent exercise before and after BA supplementation (BA: *n* = 14, control: *n* = 14). Biopsy samples were taken before and after a 28 d BA supplementation, following 4 bouts of a 30 s all-out cycling test (exercise: *n* = 7, non-exercise: *n* = 7).	Muscle carnosine levels were unaffected by exercise regardless of supplemental BA.	No change
Painelli et al. (2018) [92]	20 young healthy vegetarian men	Participants were randomly assigned to a control (*n* = 10) or high-intensity interval training (HIIT) (*n* = 10) group. HITT was performed on a cycle ergometer for 12 weeks, with progressive volume (6–12 series) and intensity (140–170% lactate threshold).	HIIT group significantly increased muscle carnosine levels compared to the control group.	Up
Hoetker et al. (2018) [97]	10 healthy, physically active men (22 ± 5 year-old)	Participants completed 2 weeks of a conditioning phase; followed by 1 week of exercise training, a single session of HITT (CPET), followed by 6 weeks of HIIT.	The levels of carnosine were increased after CPET. However, carnosine levels were decreased after 6 weeks of HIIT.	Up and down
Alabsi et al. (2023) [93]	9 male boxers (22 ± 5 year-old)	The 4-week researcher-made training program was performed three sessions per week (12 sessions in total) of around 35–45 min.	Serum carnosine levels were significantly increased by exercise, compared with pre-intervention.	Up

* Significant increase (*p* < 0.05), ** No significant change (*p* > 0.05).

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
