# Peer review of "Does Vitamin B6 Act as an Exercise Mimetic in Skeletal Muscle?"

_ijms, 2024, doi:10.3390/ijms25189962_

Round 1
Reviewer 1 Report
Comments and Suggestions for Authors
The premise of this article is that supplementation with vitamin B6 mimics the effect of exercise on skeletal muscle. Throughout the article, the authors mostly discuss studies where animals were made vitamin B6 deficient and beneficial effects on muscle were observed in comparison groups who had sufficient vitamin B6 levels. I think it is incorrect to state that taking vitamin B6 has the same effects as exercise because correcting a deficiency is quite different from seeing benefits with simple supplementation of this vitamin. I think vitamin B6 supplementation would only be beneficial if there were very large portions of the population who were deficient, but I don’t think this is the case.
Lines 74-99: Here these animal studies demonstrate the effects of a deficiency in vitamin B6, which is different than giving supplementary B6. This should be noted, as humans may not usually have this level of deficiency.
Table 1: This table is a bit deceiving if it is based on the above animal studies. The title implies these are the effects one would see with vitamin B6 supplementation. Rather, these are the effects you would see if you corrected a deficiency in vitamin B6.
Line 187: When describing this study by Suidassari et al., did they provide B6 supplementation to healthy individuals (or animals) or was this again a study where they looked at B6 deficiency and those with adequate B6 intake?
Line 250: Again, is this effect of vitamin B6 supplementation for increasing carnosine only effective in people who are vitamin B6 deficient?
Comments on the Quality of English Language
The English is fine
Reviewer 2 Report
Comments and Suggestions for Authors
In my modest opinion, this is a nice and intriguing review discussing the role of Vit. B6 on skeletal muscle and its promising support to maintain musculoskeletal health. The review is well written and the proposed tables and figure summarize the content and main findings. I have only a few minor suggestions.
Line 38: I think that it might be interesting to better define here what is mild B6 deficiency.
Lines 39-41: I would report recommended B6 intakes.
Reviewer 3 Report
Comments and Suggestions for Authors
Dear Authors,
It was a pleasure to have an opportunity to review your manuscript, which falls under a subject area as important as this. Indeed, sarcopenia is a major concern that becomes prevalent with age. Moreover, it is one of the Geriatric Giants and has serious health problems and other negative consequences, especially for older adults. On the other hand, it should be emphasized that marginal vitamin B6 deficiency is also common in various segments worldwide. Considering that increased B6 levels might be partially involved in the underlying mechanisms of the beneficial effects of exercise, it may be postulated that B6 exerts similar roles to those of exercise in improving musculoskeletal function. These observations may also facilitate the future management of sarcopenia in older adults. Finally, the manuscript is well written and the references were correctly selected and cited.
Some issues needed clarifying before publication of this paper are given below:
1. Please check your article for minor grammar issues, typos;
2. I think that terms like “the elderly” evoke negative stereotypes of older adults, which can lead to othering older adults, bias against older adults, and poor outcomes for older adults. Instead of those terms, more neutral phrases are preferred, such as “older adult, “older person,” or “persons over 65.” This term appears only once in line 266. However, please consider using for instance “older adults” instead.
Best regards,
The reviewer.
Round 2
Reviewer 1 Report
Comments and Suggestions for Authors
The authors have clarified throughout the manuscript that most research showing benefits of vitamin B6 supplementation were seen in models where there was a vitamin B6 deficiency.
That said, I think the article still strongly implies that vitamin B6 supplementation gives the same benefits as exercise training. I think this is only true in a vitamin B6 deficient state, which is not very common.
I think the manuscript would need to re-written from the perspective of changes seen if B6 supplementation was given to B6-deficient animals or individuals, without reference to how this mimics exercise training.